# Clinical Characteristics and Postoperative Complications in Patients Undergoing Colorectal Cancer Surgery with Perioperative COVID-19 Infection

**DOI:** 10.3390/cancers15194841

**Published:** 2023-10-03

**Authors:** Xuan Dai, Wenjun Ding, Yongshan He, Shiyong Huang, Yun Liu, Tingyu Wu

**Affiliations:** Department of Colorectal and Anal Surgery, Xinhua Hospital, Shanghai Jiao Tong University School of Medicine, Shanghai 200092, China; daixuan_sjtu@126.com (X.D.); wow_dingwenjun@163.com (W.D.); hys1979@yeah.net (Y.H.); huangshiyong@xinhuamed.com.cn (S.H.)

**Keywords:** colorectal cancer, COVID-19, postoperative complications, surgery, pneumonia

## Abstract

**Simple Summary:**

Compared to ancestral COVID-19 variants, Omicron variants become more infectious, but less virulent. Previous studies have recommended postponing non-emergency surgery for at least 4–8 weeks after COVID-19 infection. However, delayed surgery has been shown to be associated with tumor progression and worse overall survival for cancer patients. Here, we examined surgery risk and optimal timing for colorectal cancer patients with perioperative COVID-19 infection. A total of 211 patients who underwent colorectal cancer surgery were included. In addition, COVID-19-infected patients were further categorized into three groups based on infected time. The complication rate in patients with COVID-19 infection was 26.3%, which was significantly higher than in control patients (8.4%). Patients who underwent surgery close to the time of infection had increased surgery risks, whereas surgery performed over 1 week after recovery from COVID-19 did not increase the risk of postoperative complications. In conclusion, surgery performed during or near the time of COVID-19 infection is associated with an increased risk of developing postoperative complications. We recommend that the safe period for patients with recent COVID-19 infection in colorectal cancer surgery be at least 1 week after recovery from COVID-19.

**Abstract:**

With the emergence of novel variants, there have been widespread COVID-19 infections in the Chinese mainland recently. Compared to ancestral COVID-19 variants, Omicron variants become more infectious, but less virulent. Previous studies have recommended postponing non-emergency surgery for at least 4–8 weeks after COVID-19 infection. However, delayed surgery has been shown to be associated with tumor progression and worse overall survival for cancer patients. Here, we examined surgery risk and optimal timing for colorectal cancer patients with perioperative COVID-19 infection. A total of 211 patients who underwent colorectal cancer surgery from 1 October 2022 to 20 January 2023 at Xinhua Hospital were included. In addition, COVID-19-infected patients were further categorized into three groups based on infected time (early post-COVID-19 group, late post-COVID-19 group and postoperative COVID-19 group). The complication rate in patients with COVID-19 infection was 26.3%, which was significantly higher than in control patients (8.4%). The most common complications in COVID-19-infected patients were pneumonia, ileus and sepsis. Patients who underwent surgery close to the time of infection had increased surgery risks, whereas surgery performed over 1 week after recovery from COVID-19 did not increase the risk of postoperative complications. In conclusion, surgery performed during or near the time of COVID-19 infection is associated with an increased risk of developing postoperative complications. We recommend that the safe period for patients with recent COVID-19 infection in colorectal cancer surgery be at least 1 week after recovery from COVID-19.

## 1. Introduction

The COVID-19 pandemic has been ongoing for 3 years and remains a global concern. Patients undergoing surgical procedures are a group of individuals who are susceptible to severe acute respiratory syndrome coronavirus 2 (SARS-CoV-2) exposure and may be especially vulnerable to postoperative complications as a result of pro-inflammatory cytokine and immunosuppressive responses to the surgical intervention and mechanical ventilation [1,2]. Previous research on elective or emergent surgery on patients with COVID-19 during the perioperative period reveals a 30-day mortality rate varying from 9% to 33%, pulmonary complications from 7% to 52%, venous thrombus embolism from 7% to 13% and shock from 3% to 14%, which increase surgical risk and postoperative recovery time [1,3,4,5,6,7,8].

With the emergence of novel SARS-CoV-2 variants, the Chinese mainland had experienced a COVID-19 wave from December 2022 to January 2023, posing a critical challenge to healthcare systems. The Omicron variants, including BF.7, BA.5.2, BQ.1 and XBB, were the primary strains responsible for the current outbreak of COVID-19 in China [9,10]. Differing from ancestral SARS-CoV-2 variants, a majority of patients infected with the omicron variants in China were asymptomatic or experienced only mild symptoms [9,10]. However, little is known about the risks of surgery following infection with perioperative Omicron variants. Evidence of the safety of performing surgery on Omicron-variant-exposed patients is urgently needed. 

During COVID-19 epidemic, the infected colorectal cancer patients may face a higher surgical risk due to their compromised immune system and nutritional status; they represent a specific surgical patient population that deserves our attention. Clinical guidelines support postponing non-emergency surgery for patients with preoperative SARS-CoV-2 infection [11], but specific recommendations are inconclusive. Previous studies have recommended that the safe period for patients with COVID-19 infection be at least 4–8 weeks after SARS-CoV-2 infection [1,12,13]. However, as colorectal cancer is a progressive disease, prolonged surgical delays could pose an increased risk of tumor progression and worse overall survival. There are few data on the optimal timing for and operational risk of surgical intervention in colorectal cancer patients with COVID-19 infection. More data are urgently needed to inform clinical practice. 

Our primary objective was to examine the impact of perioperative COVID-19 (Omicron variants) infections on surgical complications in colorectal cancer. The secondary objective was to determine the optimal timing of surgery following COVID-19 infection. We hope that our findings will deepen the understanding of precise perioperative management and reduce postoperative complications for oncology surgical patients during the COVID-19 pandemic.

## 2. Materials and Methods

### 2.1. Study Population

Patients with or without perioperative COVID-19 infection who underwent colorectal cancer surgery from 1 October 2022 to 20 January 2023 at Xinhua Hospital were included in the study. The postoperative pathology of all patients was colorectal cancer, including adenocarcinoma, mucinous carcinoma and other types. Patients who underwent emergency operation were excluded. Patients who received neoadjuvant radiotherapy preoperatively were also excluded due to the potential impact of perioperative complications. The study was approved by the hospital ethics committee (XHEC-C-2023-023-1). Written informed patient consent was obtained either at the time of surgery for the use of data for scientific purposes or at follow-up when possible. 

All patients were screened before and after surgery by RT-PCR (reverse transcriptase–polymerase chain reaction assay) in nasopharyngeal swabs for SARS-CoV-2. During the study period, under rigorous COVID-19 containment protocols in China, each patient underwent SARS-CoV-2 testing every 2 days, either in the hospital or in the community. This frequent testing facilitated a comprehensive comprehension of the temporal dynamics associated with the acquisition of SARS-CoV-2, as well as the subsequent transition to a negative status. Perioperative COVID-19-infected patients were categorized into 3 groups based on the time of surgery relative to the COVID-19 test diagnosis date and turn-negative date (Figure 1). The early post-COVID-19 group was defined as surgery performed within 1 week after the COVID-19 test turned negative. The late post-COVID-19 group was composed of patients who underwent surgery between 1 and 4 weeks after the COVID-19 test turned negative. The postoperative COVID-19 group was defined as patients infected with COVID-19 within 1 week after the surgery. The control group in this study consisted of patients without COVID-19 infection. 

### 2.2. Outcomes

The primary end points of the study were postoperative complications in colorectal cancer patients with perioperative COVID-19 compared with the negative control group. The secondary objective of this study was to investigate the influence of different periods of COVID-19 infection on postoperative complications.

### 2.3. Data Collection

The following information was collected for each patient (Table 1 and Table 2): age, sex, body mass index (BMI), American Society of Anesthesiologists (ASA) class, recent smoking status, preoperative comorbidity, primary tumor site, surgical procedure, postoperative pathology, TNM staging, postoperative complication, postoperative hospital days, postoperative body temperature and laboratory findings. 

Complications were recorded according to the Clavien–Dindo classification [14] and its continuous version, the Comprehensive Complication Index (CCI) [15]. They were subdivided into anastomotic fistula, pneumonia, sepsis, urinary tract infection, Ileus, venous thrombus embolism, myocardial infarction and arrhythmia. 

The CCI was obtained by using a grading weight calculation of the Clavien–Dindo classification and then translated into an ordinal variable with 3 categories:CCI = 0: none.0 < CCI ≤ 20.9: mild.CCI > 20.9: severe.

### 2.4. Statistical Analysis

This study was performed according to STROBE guidelines for observational studies [16]. The data set included 211 patients (57 with COVID-19 and 154 controls) for 37 variables. Student’s *t*-test and Mann–Whitney U test were used for quantitative variables with normal and nonnormal distribution. Pearson’s χ^2^ test and Fisher’s exact test were used for nominal variables. A multivariable logistic regression model was used to evaluate the risk of developing postoperative complication, adjusting for covariates determined a priori to be clinically relevant. These covariates included age, sex, ASA class, BMI, recent smoking history, preoperative comorbidities (diabetes, hypertension, cerebrovascular disease, cardiovascular disease, COPD and pulmonary insufficiency) and primary tumor site. Statistical analysis was performed using SPSS 27 software. Statistical significance was defined at *p*-values < 0.05. 

## 3. Result

### 3.1. Baseline Characteristics

A total of 211 patients who had colorectal cancer surgery and met the study criteria were enrolled in this study (127 men [60.2%]; mean [SD] age, 65.6 [11.3]) (Table 1). A total of 57 (27%) patients had a perioperative COVID-19 infection, and 154 (73%) patients without COVID-19 infection were categorized as the negative control group. No significant differences were evident between the COVID-19 and control groups, considering baseline features (age, sex, BMI, ASA class, recent smoking status, preoperative comorbidity, primary tumor site, surgical procedure, postoperative pathology and TNM staging) (Table 1). 

Of the 57 perioperative infected patients, 16 (28%) patients underwent surgery within 1 week after the COVID-19 test turned negative (early post-COVID-19), 16 (28%) patients underwent surgery between 1 and 4 weeks after the COVID-19 test turned negative (late post-COVID-19) and 25 (44%) patients infected with COVID-19 within 1 week after the surgery (postoperative COVID-19) (Table 3). 

Among patients with COVID-19, 29 of 57 (50.9%) were diagnosed with rectal cancer, while the remaining were cecum and ascending colon cancer (12 [21.1%]), sigmoid colon cancer (9 [15.8%]), transverse colon cancer (4 [7%]) and descending colon cancer (3 [5.3%]) (Table 1). In total, 43 (75.4%) patients underwent laparoscopic surgery, 9 (15.8%) patients underwent open surgery and 5 (8.8%) patients underwent with robotic surgery. Postoperative pathological TNM staging was shown in the groups. There were no significant differences in these characteristics between patients with COVID-19 and controls. 

### 3.2. Postoperative Complications and Laboratory Findings

The postoperative outcomes for each group are detailed in Table 2. There was a significantly higher risk of postoperative complication in patients with perioperative COVID-19 infection compared with negative control patients (26.3% in COVID-19 patients vs. 8.4% in control patients; OR 3.87 [95% CI 1.71–8.78], *p* = 0.001), resulting in longer postoperative hospital stays (13.04 days in COVID-19 patients vs. 10.7 days in controls, *p* = 0.009). After adjustment for patient characteristics by the multivariable logistic regression model (age, sex, ASA class, BMI, recent smoking history, preoperative comorbidities (diabetes, hypertension, cerebrovascular disease, cardiovascular disease, COPD and pulmonary insufficiency) and primary tumor site), the postoperative complication rate in patients with COVID-19 was still higher compared with that of the controls (aOR 5.623 [95% CI 2.015–15.697], *p* < 0.001). To comprehensively assess various complications, the Clavien–Dindo classifications were integrated and quantified using the Comprehensive Complication Index (CCI) value. CCIs were also significantly higher in patients with COVID-19 (*p* < 0.001). 

In total, 9 of 57 (15.8%) patients with COVID-19 infection developed pneumonia after surgery, making it the most frequent complication and significantly higher compared with control patients (OR 4.62 [95% CI 1.56–13.66], *p* = 0.006). Ileus was the second most frequent postoperative adverse event (10.5%), and a significant difference was recorded compared with the control group (OR 3.5 [95% CI 1.02–11.97], *p* = 0.045). Sepsis, ranked third in frequency (7%), was also significantly higher in the COVID-19 group (OR 11.54 [95% CI 1.26–105.62], *p* = 0.03). 

To better understand the effect COVID-19 infection has on postoperative physiological changes, we also assessed various laboratory results of patients after surgery. There were numerous differences in postoperative laboratory findings between COVID-19 and control patients (Table 2), including lower lymphocyte (mean [SD] 0.66 [0.27] × 10^9^/L vs. 0.75 [0.29] × 10^9^/L, *p* = 0.048), higher C-reactive protein (median [IQR] 94 [50–137.5] mg/L vs. 70.5 [21.7–136.2] mg/L, *p* = 0.013) and higher D-dimer (median [IQR] 4.42 [3.22–8.45] mg/L vs. 3.6 [1.91–5.41] mg/L, *p* = 0.003). No differences in white blood cells, procalcitonin, cardiac troponin I, pro-brain natriuretic peptide, alanine aminotransferase, aspartate aminotransferase and creatinine were seen between the two groups. 

### 3.3. Different Periods of COVID-19 Infection on Complications

To explore the influence of viral infection at different periods on postoperative complications, 57 perioperative COVID-19-infected patients were categorized into 3 groups based on the COVID-19 test diagnosis date and turn-negative date as previously described, including early post-COVID-19 group (16/57), late post-COVID-19 group (16/57) and postoperative COVID-19 group (25/57) (Table 3 and Figure 2). 

Compared with patients who did not have COVID-19 infection, postoperative COVID-19 group patients had the highest rate of postoperative complications (36% vs. 8.4%; OR 6.1 [95% CI 2.25–16.69], *p* < 0.001) (Table 3 and Figure 2A). After adjustment for patient characteristics, postoperative group patients also had a significantly higher risk of developing complications (aOR 15.64 [95% CI 3.96–61.76], *p* < 0.001) (Figure 2B). 

Thirty-two patients had preoperative COVID-19 diagnosis. All patients were asymptomatic at the time of surgery (either never had symptoms or symptoms had resolved). There was a marginally higher risk of postoperative complications in early post-COVID-19 group patients compared with control patients by unadjusted and adjusted analysis (25% vs. 8.4%; OR 3.61 [95% CI 1.01–12.82], *p* = 0.047; aOR 5.25 [95% CI 1.15–23.9], *p* = 0.032) (Table 3 and Figure 2A,B). Notably, there was no significant difference in the postoperative complication rate between late post-COVID-19 group patients and control patients (12.5% vs. 8.4%; OR 1.54 [95% CI 0.31–7.57], *p* = 0.589; aOR 1.01 [95% CI 0.13–7.77], *p* = 0.988).

## 4. Discussion

Previous studies have indicated that infection with SARS-CoV-2 significantly increases the risk of surgery, associating it with an extremely high mortality and pneumonia complications [1,3,4,5,6,7,8]. COVID Surg Collaborative [3] has reported that postoperative pulmonary complications occur in 51.2% patients with perioperative SARS-CoV-2 infection and are associated with high mortality (23.8%). Vaccines should be prioritized for patients requiring elective surgery [17]. However, these studies were mainly based on the investigation of the ancestral SARS-CoV-2 variants. Since November 2021, Omicron, the current WHO-designated variant of concern (VOC) of SARS-CoV-2, has driven large waves of COVID-19 outbreak around the globe [18]. Compared to ancestral SARS-CoV-2 VOCs (Alpha, Beta, Gamma and Delta), Omicron has become more infectious, but less virulent [19]. The majority of individuals with Omicron infection has been reported to be asymptomatic or mild symptomatic [20]. There are still limited data on surgical patients who were infected with SARS-CoV-2 Omicron variants. 

In this cohort study, we investigated 57 colorectal cancer surgery patients with perioperative COVID-19 infection in Xinhua hospital, matched with 154 control patients with balanced demographic characteristics from 1 October 2022 to 20 January 2023. To eliminate the interference of confounding factors on the incidence of complication, a multivariable logistic regression model was used to evaluate the risk of developing postoperative complication, adjusting for covariates, including age, sex, ASA class, BMI, recent smoking history, preoperative comorbidities and primary tumor site. 

The complication rate in patients with COVID-19 infection was 26.3%, which was significantly higher than in control patients (8.4%). However, COVID-19 infection did not increase the risk of mortality, as none of the infected patients had suffered postoperative death. In particular, 15.8% of patients with COVID-19 infection had developed pneumonia, which was the most frequent complication and significantly higher compared with control patients. Despite this, the incidence rate of pneumonia was much lower than previously reported COVID-19 studies (37.5–60%) [3,4,5,6,7,8,21]. In addition, no patients in this study developed acute respiratory distress syndrome (ARDS) with their COVID-19 infection. Interestingly, ileus, which was ranked second in frequency (10.5%), was also significantly associated with COVID-19. Compared to ancestral SARS-CoV-2 VOCs, Omicrons are less likely to infect the lung or cause severe disease [20], but the impact of the Omicrons on the gastrointestinal system needs further exploration. 

Additionally, we also assessed the impact of timing of surgery relative to SARS-CoV-2 infection on the risk of developing postoperative complications. The results showed that patients who underwent surgery close to the time of infection, including early post-COVID-19 group and postoperative COVID-19 group, had increased risks of developing postoperative pneumonia, Ileus and sepsis. In particular, postoperative outcomes in postoperative COVID-19 group patients are substantially worse than all patient subgroups. Therefore, the proper preoperative evaluation, optimization of comorbidities and infection control measures should be performed help reduce the risk of complications and improve outcomes for these patients [22]. 

Previous studies have recommended that the safe period for patients with recent SARS-CoV-2 infection in whom nonemergent surgery is indicated should be at least 4–8 weeks after the SARS-CoV-2 infection [1,12,13]. However, given the apparent reduction in virulence of the Omicron variants (e.g., BF.7, BA.5.2, BQ.1 and XBB), we believe that a surgical waiting period of 4–8 weeks is unduly protracted for patients suffering from colorectal cancer. It is important to note that, for cancer patients, delayed surgical treatment has been shown to be associated with tumor progression and worse overall survival when compared to timely surgical treatment. As such, the increased surgery risks associated with SARS-CoV-2 infection should be balanced against the risks of delaying surgery in colorectal patients. To avoid excessive delays during the recovery from the SARS-CoV-2 infection, evidence-based guidelines to safely restore surgical activity are greatly needed. In the present study, we further evaluated 32 patients recovering from SARS-CoV-2 before surgery. Of note, all the patients in the early and late post-COVID-19 group were asymptomatic at the time of surgery. Surgery performed within 1 week after the COVID-19 test turned negative was still associated with an increased risk of postoperative complications (25% vs. 8.8%, *p* = 0.032), whereas surgery performed over 1 week after recovery from COVID-19 did not increase the risk of postoperative complications (12.5% vs. 8.4%, *p* = 0.988). Our findings suggest that, for colorectal cancer patients recovering from SARS-CoV-2 infection, delaying surgery for at least 1 week may reduce the risk of developing postoperative complications. 

There are several limitations to our study. The postoperative follow-up period, which ended at the time of hospital discharge, was limited in this study, and only early outcomes could be investigated. In addition, due to the limited number of COVID-19-infected cases, the sample size of patients in this study was relatively small, which does not allow for a detailed statistical analysis of specific complications in subpopulation analyses. 

## 5. Conclusions

In conclusion, surgery performed during or near the time of SARS-CoV-2 infection is associated with an increased risk of developing postoperative complications, including postoperative pneumonia, ileus and sepsis. However, surgery risks decreased to baseline in patients who underwent surgery ≥ 1 week after their COVID-19 test turned negative. We recommend that the safe period for patients with recent SARS-CoV-2 infection in colorectal cancer surgery be at least 1 week after recovery from COVID-19.

## Figures and Tables

**Figure 1 cancers-15-04841-f001:**
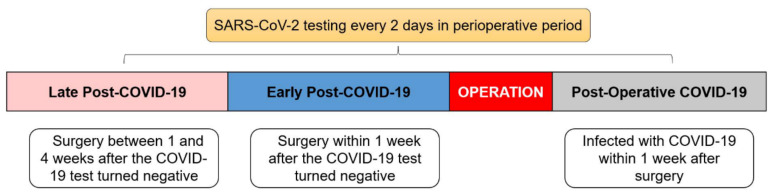
COVID-19 infection period chart. Perioperative COVID-19-infected patients were categorized into 3 groups based on the time of surgery relative to the COVID-19 test diagnosis date and turn-negative date.

**Figure 2 cancers-15-04841-f002:**
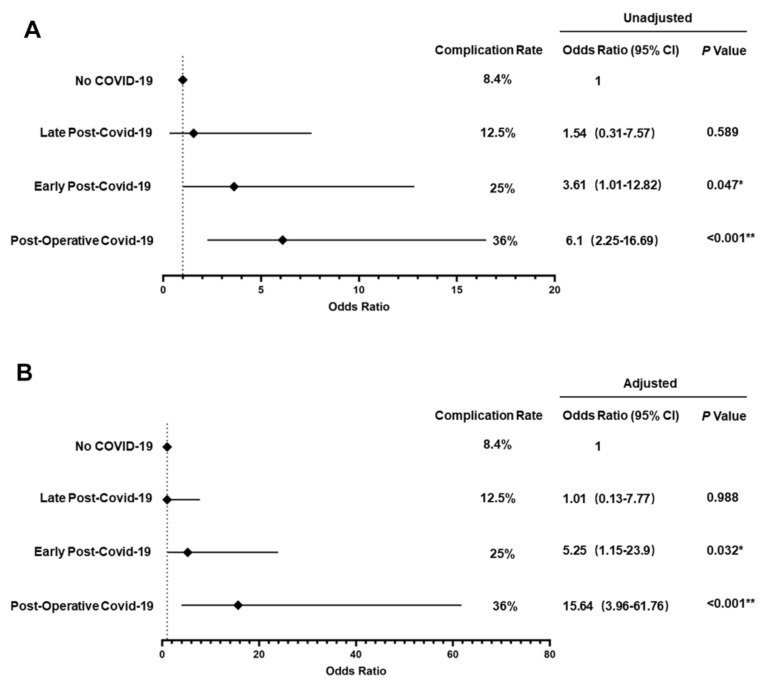
Odds ratio for complications with COVID-19 infections at different periods. Complication rates were compared between perioperative COVID-19 infection patients and control patients. (**A**) Univariable logistic regression model was used to evaluate the risk of developing postoperative complication at different COVID-19 infection periods. (**B**) Multivariable logistic regression model was used to evaluate the risk of developing postoperative complication, adjusting for different covariates, including age, sex, ASA class, BMI, recent smoking history, preoperative comorbidities (diabetes, hypertension, cerebrovascular disease, cardiovascular disease, COPD and pulmonary insufficiency) and primary tumor site. * *p* < 0.05; ** *p* < 0.01.

**Table 1 cancers-15-04841-t001:** Baseline and demographic characteristics.

	Perioperative COVID-19 Infection, No. (%)			
Variable	No (*n* = 154)	Yes (*n* = 57)	Total (*n* = 211)	*P* ^a^
Age, mean (SD), y	65.96 (10.86)	64.81 (12.54)	65.65 (11.32)	0.512 ^b^
Sex				
Male	93 (60.4)	34 (59.6)	127 (60.2)	0.922 ^c^
Female	61 (39.6)	23 (40.4)	84 (39.8)	
BMI, mean (SD)	22.07 (3.38)	22.74 (3.28)	22.25 (3.36)	0.202 ^b^
ASA Class				0.456 ^c^
I	65 (42.2)	18 (31.6)	83 (39.3)	
II	45 (29.2)	22 (38.6)	67 (31.8)	
III	34 (22.1)	12 (21.1)	46 (21.8)	
IV	10 (6.5)	5 (8.8)	15 (7.1)	
Current smoker within 1 y	21 (13.6)	7 (12.3)	28 (13.3)	0.826 ^c^
Preoperative Comorbidities				
Diabetes	21 (13.6)	9 (15.8)	30 (14.2)	0.825 ^c^
Hypertension	63 (40.9)	24 (42.1)	87 (41.2)	0.875 ^c^
Cerebrovascular disease	12 (7.8)	2 (3.5)	14 (6.6)	0.361 ^d^
Cardiovascular disease	13 (8.4)	5 (8.8)	18 (8.5)	1.000 ^d^
COPD	31 (20.1)	16 (28.1)	47 (22.3)	0.264 ^c^
Pulmonary insufficiency	96 (62.3)	33 (57.9)	129 (61.1)	0.634 ^c^
Primary Tumor Site				0.457^c^
Cecum and ascending colon	29 (18.8)	12 (21.1)	41 (19.4)	
Transverse colon	4 (2.6)	4 (7)	8 (3.8)	
Descending colon	14 (9.1)	3 (5.3)	17 (8.1)	
Sigmoid colon	19 (12.3)	9 (15.8)	28 (13.3)	
Rectum	88 (57.1)	29 (50.9)	117 (55.5)	
Operation				0.166 ^c^
Open surgery	11 (7.1)	9 (15.8)	20 (9.5)	
Laparoscopic surgery	126 (81.8)	43 (75.4)	169 (80.1)	
Robotic surgery	17 (11)	5 (8.8)	22 (10.4)	
Pathology				0.161 ^d^
Adenocarcinoma	114 (74)	49 (86)	163 (77.3)	
Mucinous carcinoma	37 (24)	8 (14)	45 (21.3)	
Other	3 (1.9)	0 (0)	3 (1.4)	
TNM staging				0.831 ^c^
I	38 (24.7)	15 (26.3)	53 (25.1)	
II	49 (31.8)	21 (36.8)	70 (33.2)	
III	49 (31.8)	16 (28.1)	65 (30.8)	
IV	18 (11.7)	5 (8.8)	23 (10.9)	

Abbreviations: COVID-19, 2019 novel coronavirus disease; SD, standard deviation; y, year; BMI, Body Mass Index; ASA, American Society of Anesthesiologists; COPD, Chronic Obstructive Pulmonary Disease. ^a^ The *p*-values indicate differences between perioperative COVID-19 negative and positive patients. A *p* < 0.05 was considered statistically significant. ^b^ Student’s *t*-test. ^c^ Pearson’s χ^2^ test. ^d^ Fisher’s exact test.

**Table 2 cancers-15-04841-t002:** Outcomes of postoperative patients.

	Perioperative COVID-19 Infection, No. (%)			
Variable	No (*n* = 154)	Yes (*n* = 57)	Total (*n* = 211)	*P* ^a^
Any Postoperative Complication				<0.001 **^b^
No	141 (91.6)	42 (73.7)	183 (86.7)	
Yes	13 (8.4)	15 (26.3)	28 (13.3)	
CCI				<0.001 **^c^
None	144 (93.5)	42 (73.7)	186 (88.2)	
Mild	3 (1.9)	8 (14)	11 (5.2)	
Severe	7 (4.5)	7 (12.3)	14 (6.6)	
Postoperative Hospital Days Mean (SD), days	10.70 (2.69)	13.04 (5.04)	11.33 (3.62)	0.007 **^d^
Complications				
Anastomotic fistula	2 (1.3)	1 (1.8)	3 (1.4)	0.613 ^c^
Pneumonia	6 (3.9)	9 (15.8)	15 (7.1)	0.005 **^c^
Sepsis	1 (0.6)	4 (7)	5 (2.4)	0.02 *^c^
Urinary tract infection	1 (0.6)	1 (1.8)	2 (0.9)	0.468 ^c^
Ileus	5 (3.2)	6 (10.5)	11 (5.2)	0.035 *^b^
VTE	1 (0.6)	1 (1.8)	2 (0.9)	0.468 ^c^
Myocardial infarction	1 (0.6)	0 (0)	1 (0.5)	1.000 ^c^
Arrhythmia	3 (1.9)	1 (1.8)	4 (1.9)	1.000 ^c^
Body temperature ^d^, mean (SD)	37.43 (0.72)	37.74 (0.82)	37.51 (0.76)	0.009 **^f^
Laboratory Findings				
WBC^d^, 10^9^/L, mean (SD)	12.48 (4.00)	12.40 (4.28)	12.46 (4.07)	0.623 ^f^
Lymph ^e^, 10^9^/L, mean (SD)	0.75 (0.29)	0.66 (0.27)	0.73 (0.29)	0.048 *^g^
CRP ^d^, mg/L, median [IQR]	70.5 [21.7–136.2]	94 [50–137.5]	77 [34–136]	0.013 *^g^
PCT ^d^, ng/mL, median [IQR]	0.19 [0.08–0.56]	0.17 [0.07–0.71]	0.19 [0.08–0.58]	0.988 ^g^
TnI ^d^, ng/mL, median [IQR]	0.008 [0.004–0.014]	0.007 [0.003–0.013]	0.008 [0.004–0.014]	0.205 ^g^
proBNP^d^, pg/mL, median [IQR]	512 [231–903]	302 [164–703]	372 [201–754]	0.288 ^g^
D-dimer^d^, mg/L, median [IQR]	3.6 [1.91–5.41]	4.42 [3.22–8.45]	3.92 [2.16–5.74]	0.003 **^g^
ALT ^d^, U/L, median [IQR]	17.5 [12–33.2]	20 [13–34]	18 [12–33]	0.214 ^g^
AST ^d^, U/L, median [IQR]	30 [23–42]	29 [22.5–42]	30 [23–42]	0.938 ^g^
Cre ^d^, umol/L, median [IQR]	65.6 [56.5–74.1]	67 [55.8–79.7]	66 [56.4–75.9]	0.672 ^g^

Abbreviations: CCI, Comprehensive Complication Index—none (CCI = 0), mild (0 < CCI ≤ 20.9), and severe (CCI > 20.9). VTE, venous thrombus embolism; WBC, white blood cell; Lymph, lymphocyte; CRP, C-reactive protein; PCT, procalcitonin; TnI, cardiac troponin I; proBNP, pro-brain natriuretic peptide; ALT, alanine aminotransferase; AST, aspartate aminotransferase; Cre, creatinine; IQR, interquartile range [25–75%]. ^a^ The *p*-values indicate differences between perioperative COVID-19 negative and positive patients. A *p* < 0.05 was considered statistically significant. ^b^ Pearson’s χ^2^ test. ^c^ Fisher’s exact test. ^d^ Maximum postoperative test results. ^e^ Minimum postoperative test results. ^f^ Student’s *t*-test. ^g^ Mann–Whitney U test. * *p* < 0.05; ** *p* < 0.01.

**Table 3 cancers-15-04841-t003:** Postoperative outcomes of patients with COVID-19 infections at different times.

	Perioperative COVID-19 Infection, No. (%)				
Variable	No COVID-19 (*n* = 154)	Late Post- COVID-19 (*n* = 16)	Early Post-COVID-19 (*n* = 16)	Postoperative COVID-19 (*n* = 25)	*P* ^a^
Any Complication					0.001 **^b^
No	141 (91.6)	14 (87.5)	12 (75)	16 (64)	
Yes	13 (8.4)	2 (12.5)	4 (25)	9 (36)	
CCI					<0.001 **^b^
None	144 (93.5)	14 (87.5)	12 (75)	16 (64)	
Mild	3 (1.9)	1 (6.3)	3 (18.8)	4 (16)	
Severe	7 (4.5)	1 (6.3)	1 (6.3)	5 (20)	

Abbreviations: CCI: Comprehensive Complication Index, none (CCI = 0), mild (8.7 ≤ CCI ≤ 20.9), and severe (CCI > 20.9). ^a^ The *p*-values indicate differences between groups. A *p* < 0.05 was considered statistically significant. ^b^ Fisher’s precise test.; ** *p* < 0.01.

## Data Availability

All data supporting the findings of this study are available within the article and from the corresponding author upon request.

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
