# Peer review of "Clinical Characteristics and Postoperative Complications in Patients Undergoing Colorectal Cancer Surgery with Perioperative COVID-19 Infection"

_cancers, 2023, doi:10.3390/cancers15194841_

Round 1

Reviewer 1 Report

Despite being a well conducted study with obvious limitations due to minimal follow-up (no data after discharge) the authors have demonstrated the risks associated with COVID infections in patients undergoing colorectal surgery. 

Several collaborative studies (with extremely large cohorts and longer follow up) have been published (Anesthesia journal and BJS) on the subject and have not been cited. 

The statistical analysis is fairly good. 

Due to the lack of novelty, it may be better to orientate the focus of the manuscript on the specific types of complications related to the Omnicron variant and colorectal surgery in the respective population. 

Author Response

Thank you for your comments and advice. Those comments are all valuable and very helpful for revising and improving our paper. Revised portion are marked in red in the paper.

1)Citations to the collaborative studies have been integrated into the discussion section. (Discussion, first paragraph).

2)  In order to make the study more innovative and to differentiate it from previous larger cohorts and longer follow up studies, our study prominently focuses on Omicron variant and colorectal surgery, as emphasized in the manuscript.

Reviewer 2 Report

The paper has assessed the risk of post-operative complications in patients who contract the Omicron variant of COVID-19 virus.  Since only a part of these patients experience complications, one mighy ask whether vaccination against COVID-19 virus performed in the months preceding the operation can play a role in avoiding complications. Did have the Authors evaluated this factor in their patients? 

Author Response

Thank you for your comments. Unfortunately, our study did not evaluate whether patients underwent preoperative vaccination, which is indeed a question worthy of investigation. However, it is essential to note that in China, the vast majority of the population has been vaccinated, and unvaccinated individuals are rare. Therefore, it is challenging to study the influence of vaccination on postoperative complications.

Reviewer 3 Report

Thank you for this opportunity.

I read your study. The idea is very good, and the results are sound and interesting.

However, my main question is if the study can say what they find.

In other words, 57 COVID + in total subdivided into three groups are sufficient power to do that?

The English and ideas are good, but the references are few.

Can you look at this for the organization? 

Learning from the Italian experience during COVID-19 pandemic waves: be prepared and mind some crucial aspects. Acta Biomed. 2021 May 12;92(2):e2021097. doi: 10.23750/abm.v92i2.11159. PMID: 33988143; PMCID: PMC8182622.

Best Regards

Good

Author Response

(1) We are extremely grateful to reviewer for pointing out the limitation of the small sample size of this study, we have discussed the limitations of this study in the discussion section. Although the number of cases is limited, the statistical analysis in this study was conducted using the chi-square test and Fisher exact test. Due to the significant variation in different sub-groups, we observed the difference in complication rate holds statistical significance. However, the results of this study were preliminary. Prospective validations in large cohorts are required.

(2) This paper (Cristian Deana, et al.) also elucidates the necessity to postpone elective surgery, and we have cited it in the introduction section(Introduction, third paragraph).